# Early-Life Antibiotic Exposure and Childhood Asthma Trajectories: A National Population-Based Birth Cohort

**DOI:** 10.3390/antibiotics12020314

**Published:** 2023-02-03

**Authors:** Yankun Lu, Yichao Wang, Jing Wang, Adrian J. Lowe, Luke E. Grzeskowiak, Yanhong J. Hu

**Affiliations:** 1Murdoch Children’s Research Institute, The Royal Children’s Hospital, Parkville, VIC 3052, Australia; 2Melbourne School of Population and Global Health, The University of Melbourne, Parkville, VIC 3053, Australia; 3Centre for Social and Early Emotional Development, School of Psychology, Deakin University, Geelong, VIC 3220, Australia; 4Department of Pediatrics, The University of Melbourne, Parkville, VIC 3010, Australia; 5Allergy and Lung Health Unit, Centre for Epidemiology and Biostatistics, Melbourne School of Population and Global Health, The University of Melbourne, Parkville, VIC 3053, Australia; 6College of Medicine & Public Health, Flinders University, Adelaide, SA 5042, Australia; 7SAHMRI Women and Kids, South Australian Health and Medical Research Institute, Adelaide, SA 5000, Australia

**Keywords:** early-life, antibiotic, childhood, asthma, trajectory, birth cohort

## Abstract

Introduction: Early-life antibiotic exposure is common and impacts the development of the child’s microbiome and immune system. Information on the impacts of early-life antibiotics exposure on childhood asthma is lacking. Methods: This study examined associations between early-life (0–24 months) antibiotics exposure with childhood (6–15 years) asthma trajectories through the Australian Longitudinal Study of Australian Children (LSAC) and their linked data from the Pharmaceutical Benefits Scheme. Asthma phenotypes were derived by group-based trajectory modeling. Results: Of 5107 LSAC participants, 4318 were included in the final analyses (84.6% retention). Four asthma phenotypes were identified: Always-low-risk (79.0%), early-resolving asthma (7.1%), early-persistent asthma (7.9%), and late-onset asthma (6.0%). Any early-life antibiotic exposure increased risk 2.3-fold (95% CI: 1.47–3.67; *p* < 0.001) for early-persistent asthma among all children. In subgroup analyses, early-persistent asthma risk increased by 2.7-fold with any second-generation cephalosporin exposure, and by 2-fold with any β-lactam other than cephalosporin or macrolide exposure. Conclusion: We concluded that early-life antibiotic exposure is associated with an increased risk of early-persistent childhood asthma. This reinforces scrutiny of early-life antibiotic use, particularly for common viral infections where no antibiotics are required.

## 1. Introduction

Asthma is the most common childhood respiratory disease in Australia, with AUD 770 million in related medical costs per year, imposing a health and economic burden on families and society [1]. However, the exact cause of asthma is unknown. Common asthma triggers include infections, allergies, smoke, pollution, and medicines. One study from Sweden found that more than two-thirds of children received antibiotics before the age of two years [2]. This is of concern because antibiotic utilization has been demonstrated to contribute to antibiotic resistance, while also being associated with altered immune maturation, neurodevelopmental disorders, atopic diseases, and metabolic disorders in childhood [3]. Antibiotics alter children’s gut microbiota diversity and may actually increase susceptibility to infections [3], potentially contributing to asthma pathogenesis.

There is a range of evidence to support the possibility that early-life antibiotics may increase the risk of asthma. Ecological and secular trend evidence indicate that the rates of both antibiotic use and asthma prevalence have risen in parallel, although children were only followed until age 5 years and potential unadjusted confounders [4,5]. Several cohort studies have now shown that antibiotic exposure under the age of 1 year is associated with an increased risk of childhood asthma up to age 4 years [6,7,8,9], especially broad-spectrum antibiotic classes [4,8]. Previous studies have usually investigated asthma as an outcome at a single time point [5,6,8,9,10], which fails to consider the dynamic nature of wheezy disorders during early childhood. The majority (65%) of early-life wheezing resolves by 6 years of age [11,12] and there is difficulty identifying which early wheeze patterns represent an asthma phenotype [13]. While it is now well established that different asthma trajectories across the whole of childhood may have different underpinnings and future adult health outcomes [9,14,15,16,17,18], no studies so far have examined whether early-life antibiotic exposure may contribute to longitudinal asthma phenotypes.

The Australian Pharmaceutical Benefits Scheme (PBS) [19], holds data on outpatient prescriptions which can be linked to cohort participant dispensing data. This provides an opportunity to address the knowledge gaps. This study used data from the Longitudinal Study of Australian Children (LSAC), a nationally representative population study linked with dispensing-record data from PBS, which recruited and followed children between 0–15 years of age. This study investigated (1) childhood asthma trajectories, and (2) the associations between early-life antibiotic exposure and childhood asthma phenotypes after adequate confounder adjustments.

## 2. Materials and Methods

### 2.1. Study Design and Participants

LSAC, Australia’s only nationally-representative children’s cohort study, focuses on answering research questions around parenting, family relationships, education, childcare, employment, and health [20]. In this study, data from waves 1–8 of the LSAC study ‘B’ cohort were used as this included the infant period up to childhood, enabling information on early-life antibiotic exposure and childhood asthma to be assessed. The B cohort (*n* = 5107, 50% uptake) recruited Australian-born children between 2003–2004 (wave 1). In wave 1, 93% of LSAC children’s Medicare PBS-dispensing data were linked [20]. Sample size decreased in every consecutive wave due to attrition and by wave 8 (2018), there were 3127 (retention rate = 61%) children remaining in the B cohort (Figure 1).

#### Procedures

Detailed procedures have previously been described [21]. In brief, following consent, biennial face-to-face interviews were conducted by trained interviewers with children and primary caregivers (97% biological mothers) in the home. From wave 4, computer-assisted child-self-report was introduced for use by the child and caregiver. If parents lived elsewhere, consent was obtained before completing a leave-behind questionnaire.

LSAC was approved by the Australian Institute of Family Studies Ethics committee [20]. Written consent was obtained from parents/guardians for their own and their child’s participation.

### 2.2. Measures

#### 2.2.1. Exposure (Antibiotic Exposure in Early-Life up to 24 Months)

Exposure to antibiotics from birth to 24 months was identified through linked PBS data, which recorded the type, supply date, and frequency of use (i.e., 0, 1, 2, 3, ≥4 courses) at the point of pharmacist dispensing. PBS only recorded antibiotics dispensed with a patient co-payment greater than $23.70 in 2004 (wave 1), except for those parents with a healthcare card, which entitles them to cheaper medicines [22,23]. Healthcare cards are issued to those eligible based on health and economic needs [24]. Among LSAC families, 22.4% had healthcare cards and more than 60% of those families are of low socioeconomic status [25]. The most commonly prescribed drug in Australia in 2004 was amoxicillin (age not specified) [26]. A study of children 0–12 years old in 2013 found that amoxicillin was the most used antibiotic [27]. The average price for amoxicillin was $10.6 in 2004 [28]. Children whose parents have healthcare cards were included in a restricted analysis for a comprehensive assessment of antibiotics exposure among this group.

All antibiotics were categorized according to the Australian Therapeutic Goods Administration [29], which includes *β*-lactam other than cephalosporins, cephalosporins (1st–4th generations), quinolone, macrolide, glycopeptide, tetracycline, lincosamide, aminoglycoside, sulfonamide, nitroimidazole, fusidane, methenamine, and trimethoprim. All included antibiotics were oral or injectable. Topical use including eye/ear drops and skin cream/ointment was excluded.

#### 2.2.2. Outcome (Asthma Trajectories in Children)

At each wave from waves 5–8, parents reported their child’s current asthma status, based on a doctor’s diagnosis (Has your doctor ever told you that study child has asthma?); plus either a past 12-month asthma medication history (Has study child taken any medication for asthma in the last 12 months?) or, symptoms of wheezing in the past 12-months (In the last 12 months, has study child ever had an illness with wheezing in the chest which lasted for 1 week or more?) (Appendix A). This definition has been adopted by several previous studies [5,8,9]. Only children with current asthma for at least two waves were included in the trajectory analysis. We included children above 6 years (from wave 4) in the trajectory analysis to exclude the potential for asthma misdiagnoses in young children [30].

#### 2.2.3. Confounders

Potential confounders were identified a priori through relevant literature [31,32,33] and a directed acyclic graph (Appendix A), with codes provided in Appendix A. Identified confounders (Appendix A) included: (1) mother-related factors: maternal asthma; smoking during pregnancy; antibiotic exposure during pregnancy, ethnicity; education; mode of delivery; breastfeeding; rurality (urban or rural); Socio-Economic Indexes for Areas (SEIFA, a ranking system for socioeconomic position as quintiles (1–5)); (2) child-related factors: sex; preterm birth; low birth weight; childhood obesity, potential infections which indicate antibiotic use (hospitalization for fever/viral infection at age 0–1); ear infection or other infectious condition (other than diarrhea/colitis, ear infection, and other illnesses) at age 0–1 year; (3) family-related factors: children in household; presence of older sibling in household.

### 2.3. Statistical Analysis

All statistical analyses were performed in Stata 17.0 (StataCorp LLC, College Station, TX, USA, 2021). Descriptive analysis was used to identify the characteristics of the study population. For asthma trajectories, we used group-based trajectory modeling (GBTM) by the ‘traj’ plug-in in Stata [34]. GBTM is a statistical method that identifies groups of separate trajectories that fall under the same polynomial functions as per maximum likelihood estimation [35]. Models were selected based on several goodness-of-fit indicators (Appendix A), such as Akaike’s information criterion, Bayesian information criterion, meaningful shapes, and an average posterior probability >0.7 as recommended by Nagin and Odgers [36] (Appendix A). Multinomial logistic regression and adjusted risk ratio (aRR) were used to assess the relationship between antibiotic exposure and asthma trajectories. We selected the always-low-risk asthma group as the reference group and adjusted for confounders in a multinomial regression model. We performed three additional subgroup analyses: (1) sex, (2) antibiotic exposure in early life during 0–6 months, 6–12 months, and 12–24 months, and (3) different antibiotic class. Restricted analysis was used for children whose parents have healthcare cards. Interactions between sex, antibiotic-exposure periods, and trajectory were tested using the likelihood ratio test (*p*-value < 0.05).

## 3. Results

### 3.1. Sample Characteristics and Antibiotic Exposure

Of 5107 total children, 4318 were included in our analysis (Figure 1). Of these, 1028 (32.7%) had asthma ever reported. Table 1 shows the baseline subject characteristics. The mean maternal age was 30.8 ± 5.5 years. During pregnancy, 185 (3.6%) mothers used asthma medication, 531 (10%) mothers used antibiotics and 709 (14%) smoked. The mean birthweight of the children was 3395 ± 610 g, 2608 (51%) were males, 2845 (75%) were not exclusively breastfed before 6 months of age, and 1503 (29.5%) were delivered by Cesarean section.

In the PBS dataset, 1267 (24.8%) children received at least one antibiotic course during their first 24 months of life (Table 2). The number of children whose parents have healthcare cards exposed to at least one antibiotic course was 778 (68.1%). Early-life antibiotic exposures were as either a single course for 317 (25.0%) children or two or more courses for 950 (75.0%). The most exposed antibiotic was β-lactam other than cephalosporins with 1123 (88.6%) children, followed by second-generation cephalosporin (351 (27.7%)) and first-generation cephalosporin (257 (20.3%)), macrolide (252 (19.7%)), and sulfonamide/trimethoprim (98 (7.7%)). Antibiotics prescribed for children whose parents with healthcare cards followed a similar trend, with β-lactam other than cephalosporins (688 (88.4%)) ranking first, followed by second-generation cephalosporins (224 (28.8%)) (Appendix A).

### 3.2. Childhood Asthma Trajectory

Asthma outcomes from 6–15 years followed four distinct trajectory groups (Figure 2). The “always-low-risk” group (3508 (79.0%)), with a consistently low probability of asthma, which served as the reference group for subsequent analyses. Children in the “early-resolving” group (213 (7.1%)) had developed asthma at 6 years old, which resolved steadily as age increased. The “early-persistent” group (384 (7.9%)) was characterized starting at age 6 with a consistently high probability of asthma up to 15 years old. The “late-onset” group (213 (6.0%)) had a low probability at 6 years old, increasing from age 6 steadily until 15 years old. Trajectory patterns were similar for girls and boys; however, trajectory shape for late-onset asthma for boys increased at a younger age and achieved a higher peak compared to girls (Appendix A).

### 3.3. Early-Life Antibiotics Exposure Association with Childhood Asthma Trajectory

#### Primary Analysis

Figure 3 illustrates early-life antibiotics exposure and its associations with asthma trajectories. For all children, after adjusting for maternal, child, and family confounders, (Appendix A), early-life antibiotics exposure was associated with a 2.1-fold (95% CI: 1.50–2.86; *p* < 0.001) increase in early-persistent asthma compared to the “always-low-risk” asthma group and no significant associations for early-resolving asthma and late-onset asthma groups. Similar association patterns were seen for children whose parents without and with a healthcare card (2.3-fold 95% CI: 1.47–3.67; *p* < 0.001 vs. 2.2-fold 95% CI: 1.02–4.66; *p* = 0.045, respectively) increase in the risk of early-persistent asthma compared to the “always-low-risk” asthma group. There were no obvious associations with the risk of early-resolving or late-onset asthma.

### 3.4. Subgroup Analyses

Early-life antibiotic exposure for both boys and girls was associated with 1.6-fold (95% CI: 1.03–2.44; *p* = 0.04) and 3.0-fold (95% CI: 1.84–5.03; *p* < 0.001) increased risk of early-persistent asthma, respectively. There was an additional association between antibiotic exposure and late-onset asthma for boys only (adjusted risk ratio: 2.0; 95% CI: 1.15–3.60; *p* = 0.02). Further interaction tests showed that sex is an effect modifier, with a higher risk of early-persistent asthma in girls and late-onset asthma in boys (Appendix A). Restricted analysis for children whose parents have healthcare cards showed early-life antibiotic exposure increased the risk of early-persistent asthma by 3.9-fold (95% CI: 1.07–14.22; *p* = 0.04) for boys compared to always-low-risk asthma (Appendix A).

Time of antibiotic exposure between 0–6 months and between 12–24 months was associated with an approximately 1.9-fold increase in the risk of early-persistent asthma, while exposure between 6–12 months was associated with a 2.6-fold risk of early-persistent asthma. Further interaction testing showed that time of exposure is an effect modifier, with antibiotic exposure between 6–12 months showing the highest risk for early-persistent asthma (Appendix A).

Among different antibiotic classes, second-generation cephalosporin exposure was associated with increased risk for early-persistent asthma (adjusted risk ratio: 2.7; 95% CI: 1.69–4.20; *p* < 0.001). Any β-lactam other than cephalosporins or macrolide use was associated with a 2-fold increased risk for early-persistent asthma. There were marginal associations between any sulfonamide and trimethoprim use and early persistent asthma (Appendix A). There was a linear relationship between the number of courses of antibiotics and asthma trajectory. For antibiotics exposure to ≥3 courses, there was a 2.6–fold risk for early-persistent asthma compared to always-low-risk asthma (Figure 4).

## 4. Discussion

### 4.1. Principal Findings

We have found that children exposed to early-life antibiotics were twice as likely to have early-persistent asthma compared to always-low-risk asthma children after adjusting for confounders. Restricting the analysis to children whose parents had healthcare cards found that early-life antibiotics exposure had a marginal association with early-persistent asthma. Antibiotic exposure between 6–12 months of age had the highest adjust risk ratios with early-persistent asthma compared to 0–6 months and 12–24 months. Exposure to β-lactam other than cephalosporins, second-generation cephalosporin, or macrolides was associated with early-persistent asthma and second-generation cephalosporin had the highest risk ratio. Using more than 3 courses of antibiotics was associated with an increased risk for early-persistent asthma compared to always-low-risk asthma.

### 4.2. Correlation with Existing Evidence

The four asthma phenotypes identified in our study are consistent with previous published studies [15,17,18] exploring asthma trajectories. Previous studies investigating asthma trajectories found a decreasing trend in early-persistent asthma between 8–12 years [16,18], while our study shows a consistently high trend for asthma from age 6–15 years old. This might be explained by variations in asthma diagnoses from different geographic locations [18] and whether a study used a broad definition including asthma, eczema, and food allergy [16]. In our analysis, the shape of asthma trajectories for boys and girls were mostly similar, which is consistent with studies from Australia [16] and Denmark [15]. However, our study showed that boys had an early age (8/9) for the risk of early-persistent asthma and achieved a peak at age 14/15 compared to girls. This correlated with the findings by Postma et al. [37], which showed dynamic changes in asthma prevalence of “sex switch”, where more boys tend to have asthma in early childhood and vice versa in adulthood. In addition, our study also showed that there were more boys in the early-resolving and early-persistent asthma groups. A potential explanation is that the smaller airway sizes found in boys under the age of 10 years than in girls [38] may lead to an increased risk of childhood asthma that usually resolves around puberty.

Our study found that early-life antibiotic exposure was associated with early-persistent asthma, providing important evidence on the development of childhood asthma. Our findings support the results of Yoshida et al., where antibiotic exposure had stronger associations with early childhood asthma than late-onset asthma [9]. Since their study population was up to 6 years old, our finding has expanded this observation to the whole of childhood. Our finding of early-life antibiotic exposure associated with a 2-fold increase in the risk of early-persistent asthma is consistent with a potentially important effect on children’s health. It is comparable to the finding from another study that showed a 1.90-fold increase in odds of current asthma at age 12 years following antibiotic consumption within the first week of life [39]. Furthermore, we observed an association between early antibiotic exposure and persistent asthma at an age of 15 years, which is also likely to persist further into adulthood [18]. While underlying mechanisms remain speculative, findings may be attributed to antibiotic-induced microbiota changes and altered immune function at a critical time during immunological development, which may lead to irreversible harm to lung function [4,18]. Further studies are required to confirm the proposed mechanisms linking early antibiotics exposure and asthma development.

Studies have suggested that the association between early-life antibiotic use and childhood asthma could be due to protopathic bias and reverse causation [40]. Protopathic bias occurs when the early signs of disease (which is yet to be diagnosed) are treated with medication [41]. This might be a potential issue in our study, particularly for the early onset persistent asthma group, as we do not have data available to exclude this possibility. Örtqvist et al. [5] have concluded in their study that the association between childhood antibiotics and asthma could be impacted by protopathic bias (a form of reverse causation), as the odds for asthma were higher for children treated with antibiotics indicated for respiratory illnesses (which may have been early onset asthma) than urinary or skin conditions. Unlike our study, in their study, early wheezing was not excluded, which could allow protopathic bias to remain. In our study, asthma was assessed from 6–15 years of age, which satisfies the two-year “wash-out” period criteria between antibiotic exposure and asthma diagnosis. Moreover, respiratory illness in the Örtqvist study was assumed by antibiotics categories, which may not reflect child conditions correctly.

Our findings provide additional insights into antibiotic class exposures and asthma development. Among all antibiotics, second-generation cephalosporins had the highest risk ratio for early-persistent asthma, perhaps because second-generation cephalosporins are more often used for young children with asthma [42], leading to potential confounding by respiratory infection rather than a causal effect. Murk et al. [40] showed that the odds drop from 1.38 to 1.16 for child antibiotics exposure and asthma but remained significant after adjusted respiratory infections, which indicated that there might be potential confounding factors like respiratory infections. However, we have adjusted for potential infections that indicated antibiotic use (hospitalization for fever/viral infection at age 0–1; ear infection or other infectious condition (other than diarrhea/colitis, ear infection, and other illnesses), suggesting likelihood bias by infections were reduced. The majority of respiratory infections experienced by young children are viral in nature and do not require antibiotic treatment. Unnecessary treatment of viral infections with antibiotics contributes to the global crisis of antibiotic resistance [43,44]. In young children, wheezing symptoms may lead to an asthma diagnosis, which may resolve with immune system maturation and airway caliber enlargement. We also found a dose-response relationship between increasing courses of antibiotics and increased risks of childhood early-persistent asthma, which is consistent with a previous study [45].

The timing of antibiotic exposure is critical. Our study showed antibiotic exposure between 6–12 months was associated with the highest increased risk of early-persistent asthma. A study by Ni et al. [46] showed first-year antibiotic exposure significantly increased lifetime asthma risk. Patrick et al. [4] suggested that antibiotic exposure during the “critical developmental window” of the infant gut may have long-standing and irreversible consequences for the microbiome, promoting immune system disorders or delayed immune maturation [47]. The exact mechanism remains unknown, but there may be a critical window between 6 and 24 months for a child’s immune system development.

### 4.3. Limitations

Our study had several imitations. First, the PBS data could potentially lead to underestimation of exposure since antibiotics costing individuals less than $23.70 at wave 1 would not be recorded in the PBS data [22], while the most dispensed antibiotic (amoxicillin) for children was less than half that price. This might lead to at least half of the antibiotics data being absent from the PBS dataset. The restricted analysis on children with healthcare cards provides a fuller picture of antibiotic exposure, which is limited to a smaller, socioeconomically disadvantaged sample in the trajectory, but it maintained the same trends and patterns after adjusting for socioeconomic status. Second, possible confounding by indication may occur as there was no available data for antibiotics prescribing indication. However, we adjusted for parent-reported early infections (e.g., ear infections, hospitalization for fever/viral infection, and other infections). These infections themselves may have underlying influences on childhood asthma, which could serve as a residual confounder that was not completely adjusted for in this study. Finally, antibiotics exposure may not represent consumption, and asthma outcomes might have potential recall bias. However, studies have found that most parents usually have a high adherence rate for giving infant and child medications compared to their own [48]. Parents are also better at remembering their children’s condition and treatment.

## 5. Conclusions

Within the LSAC B cohort, we have identified four childhood asthma trajectories including always-low-risk, early-resolving, early-persistent, and late-onset asthma. Antibiotic exposure between 0–24 months and using more than three courses were both associated with increased risk for childhood early-persistent asthma. Our findings emphasize the importance of antibiotic stewardship and greater caution relating to the widespread and potentially indiscriminate use of antibiotics in early life.

## Figures and Tables

**Figure 1 antibiotics-12-00314-f001:**
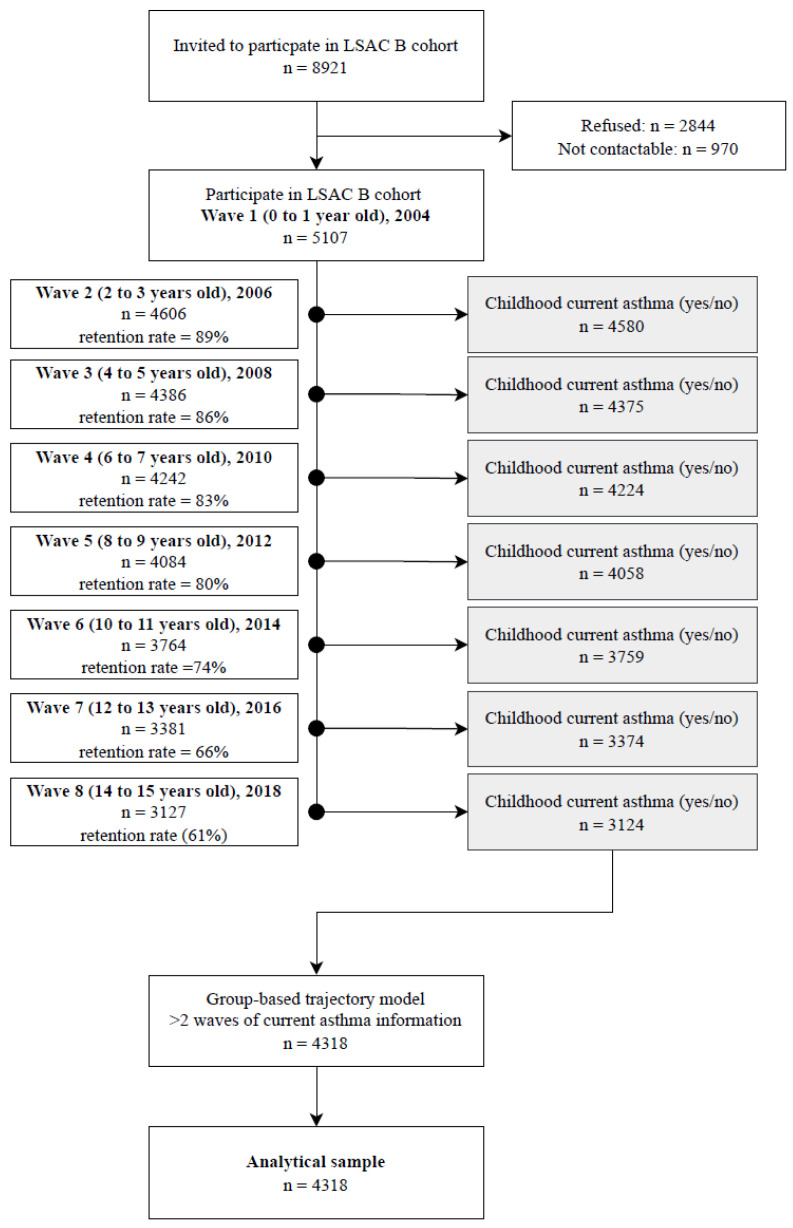
Flow chart defining the initial and final study population.

**Figure 2 antibiotics-12-00314-f002:**
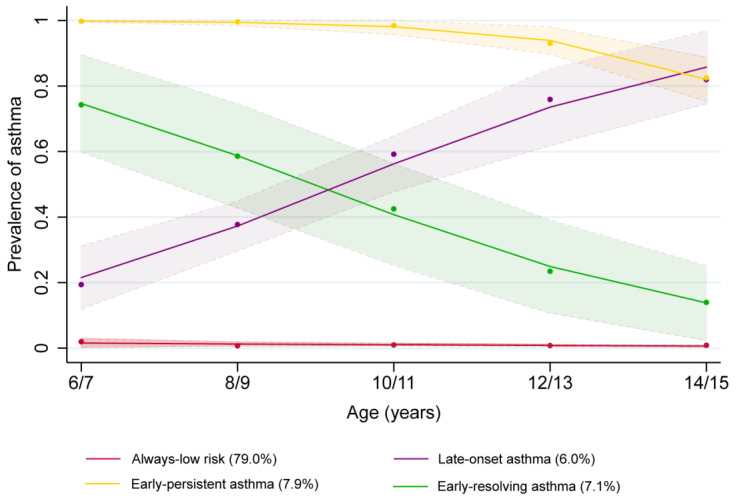
Phenotypes of asthma identified by group-based trajectory analysis using the B cohort of the Longitudinal Study of Australian Children between 2004–2018.

**Figure 3 antibiotics-12-00314-f003:**
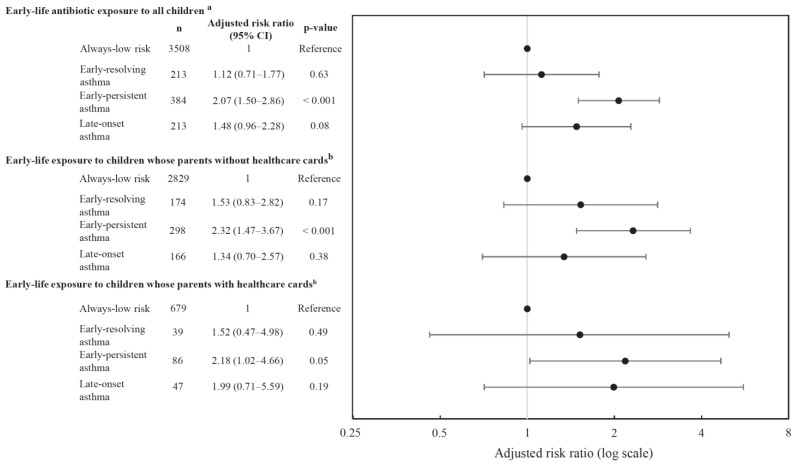
Association between early-life antibiotics and asthma phenotypes. ^a^ For confounders adjusted include maternal smoking, maternal alcohol, maternal education, ethnicity, Socio-Economic Indexes for Areas (SEIFA), low birthweight, delivery mode, preterm birth, maternal asthma, rurality, exclusive breastfeeding, viral respiratory infection, ear infection, and other infections, number of siblings in household, presence of older sibling, maternal antibiotics, childhood obesity. ^b^ Restrict analysis for parents with healthcare cards, adjustments include the variables from confounders. Abbreviations: *n* = total number of children in the trajectory group.

**Figure 4 antibiotics-12-00314-f004:**
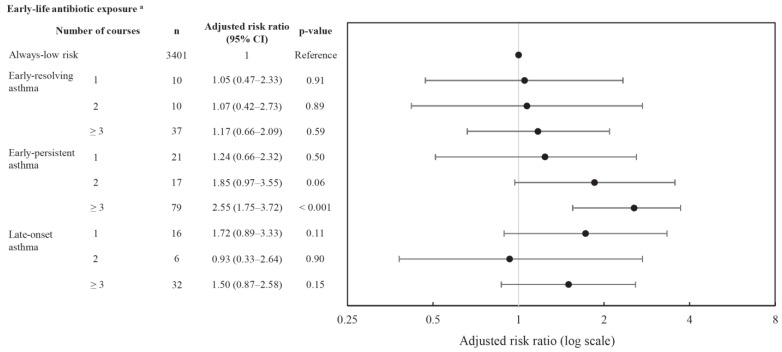
Association between early-life antibiotic exposure courses and asthma phenotypes. ^a^ For confounders adjusted include maternal smoking, maternal alcohol, maternal education, ethnicity, Socio-Economic Indexes for Areas (SEIFA), low birthweight, delivery mode, preterm birth, maternal asthma, rurality, exclusive breastfeeding, viral respiratory infection, ear infection, and other infections, number of siblings in household, presence of older sibling, maternal antibiotics, childhood obesity. Abbreviations: *n* = total number of children in the trajectory group.

**Table 1 antibiotics-12-00314-t001:** Characteristics of study population.

**Sample,** * **n** * **(%)**	**All** **Children** * **n** * **= 5107**	**Children with HCC †** * **n** * **= 1143**
Infant		
Infant birthweight (g), mean ± SD	3395.4 ± 610	3305.8 ± 694
Gestational age weeks, mean ± SD	39.3 ± 2.0	39.1 ± 2.2
Infant sex, %		
Male Female	2608 (51.1) 2499 (48.9)	590 (51.6) 553 (48.4)
Ever had Asthma, *n* (%)		
Yes No	1028 (32.7) 2117 (67.3)	220 (39.3) 340 (60.7)
Delivery method, *n* (%)		
Vaginal Breech Cesarean section Vacuum extraction Forceps Others Unknown	3146 (61.7) 24 (0.5) 1503 (29.5) 248 (4.9) 161 (3.2) 21 (0.4) 4 (0.08)	760 (66.7) 3 (0.3) 306 (26.8) 35 (3.1) 33 (2.9) 3 (0.3)
Feeding method at 6 months, *n* (%)		
Full breastfed exclusive Formula-fed	944 (24.9) 2845 (75.1)	153 (17.4) 725 (82.6)
Ever been hospitalized for respiratory infection by 24 months, *n* (%)		
Yes No	111 (2.6) 4975 (97.4)	34 (3.0) 1101 (97.0)
Ever had ear infection by 12 months, *n* (%)		
Yes No	200 (3.9) 4883 (96.1)	50 (4.4) 1088 (95.6)
Residence location, *n* (%)		
Urban Rural	4633 (90.9) 462 (9.1)	1022 (90.0) 155 (10.0)
Birthweight (g), *n* (%)		
<2500 (low birthweight) 2500–2999 3000–3999 4000–5499	294 (5.8) 706 (13.9) 3438 (67.4) 656 (12.9)	93 (8.2) 182 (16.0) 734 (64.4) 130 (11.4)
Preterm birth, *n* (%)		
Yes No	334 (6.5) 4773 (93.5)	93 (8.1) 1050 (91.9)
Ethnicity, *n* (%)		
Australian Non-Australian	3082 (60.3) 2025 (39.7)	537 (47.0) 606 (53.0)
Maternal		
Maternal age (years), mean ± SD	30.8 ± 5.5	28.8 ± 6.7
Quintiles of neighborhood disadvantage (SEIFA), *n* (%)		
1 (most disadvantaged) 2 3 4 5 (least disadvantaged)	1136 (22.2) 1101 (21.6) 1038 (20.3) 863 (16.9) 969 (19.0)	359 (31.4) 306 (36.8) 222 (19.4) 145 (12.7) 111 (9.7)
Maternal asthma, *n* (%)		
Yes No	185 (3.6) 4912 (96.4)	47 (4.1) 1093 (95.9)
Maternal smoking, *n* (%)		
Yes No	709 (16.7) 3530 (83.3)	279 (32.6) 576 (67.4)
Maternal alcohol		
Yes No	1633 (38.6) 2594 (61.4)	216 (25.5) 631 (74.5)
Mother’s highest education achieved at conception, *n* (%)		
Less than year 12 Completed year 12 Bachelor’s degree, advanced diploma, diploma, or certificate Postgraduate degree, graduate diploma, or certificate	865 (16.9) 796 (15.6) 2764 (54.2) 679 (13.3)	348 (30.5) 181 (15.8) 541 (47.3) 73 (6.4)
Antibiotic exposure during pregnancy, *n* (%)		
Yes No	531 (10.4) 4566 (89.6)	138 (12.1) 1002 (87.9)

Note: Abbreviations: SD = Standard deviation, g = gram(s); *n* = total number of children in the model; SD = standard deviation; SEIFA = Socio-Economic Indexes for Areas. Children with HCC † = Children whose parents with healthcare cards.

**Table 2 antibiotics-12-00314-t002:** Early-life antibiotics exposure up to 24 months in LSAC B cohort, Australia.

**Any Antibiotic Prescribed (PBS Records), ** * **n** * **(%) †**	
Yes No	1267 (24.8) 3840 (75.2)
Any antibiotic prescribed for children whose parents have healthcare cards, *n* (%)	3395.4 ± 610
Yes No	778 (68.1) 365 (31.9)
Number of antibiotic courses exposed during the period, *n* (%)	
1 2 ≥3	317 (25.0) 233 (18.4) 717 (56.6)
**Class of Antibiotics Exposed by Courses, *n* (%)**	
β-lactam other than cephalosporin ‡, total	1123 (100)
1 2 3 ≥4	382 (34.0) 254 (22.6) 153 (13.6) 334 (29.8)
First-generation cephalosporin, total	257 (100)
1 2 3 ≥4	165 (64.2) 50 (19.5) 25 (9.7) 17 (6.6)
Second-generation cephalosporin, total	351 (100)
1 2 3 ≥4	185 (52.7) 83 (23.6) 37 (10.5) 46 (13.1)
Third-generation cephalosporin, total	3 (100)
1	3 (100)
Quinolone, total	2 (100)
1 2	1 (50) 1 (50)
Macrolide, total	252 (100)
1 2 3 ≥4	167 (66.3) 52 (20.6) 16 (6.4) 17(6.7)
Aminoglycoside, total	1 (100)
1	1 (100)
Sulfonamide and Trimethoprim, total	98 (100)
1 2 3 ≥4	71 (72.5) 13 (13.3) 7 (7.1) 7 (7.1)
Nitroimidazole, total	30 (100)
1 2	28 (93.3) 2 (6.7)
Fusidane, total	1 (100)
1	1 (100)

Abbreviations: *n* = total number of children in the model; PBS = Pharmaceutical Benefit Scheme; SD = standard deviation; SEIFA = Socio-Economic Indexes for Areas; † Percentage = number of antibiotic exposures/the total number of antibiotic exposures in that class × 100%; ‡ β-lactam other than cephalosporin = ticarcillin, penicillin, phenoxymethylpenicillin, procaine penicillin, flucloxacillin, dicloxacillin, benzylpenicillin, ampicillin, amoxicillin, amoxicillin/clavulanic acid.

## Data Availability

The data that support the findings of this study are available from the Australian Data Archive (ADA) but restrictions apply to the availability of these data, which were used under license for the current study, and so are not publicly available. Data are however available from the authors upon reasonable request and with permission from the Australian Data Archive and the corresponding author. The corresponding author, Hu Y. J, had full access to all the data in the study and had final responsibility for the decision to submit for publication.

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
