# Peer review of "Early-Life Antibiotic Exposure and Childhood Asthma Trajectories: A National Population-Based Birth Cohort"

_antibiotics, 2023, doi:10.3390/antibiotics12020314_

Round 1
Reviewer 1 Report
It is an interesting study highlighting that antibiotic exposure between 0-24 months is associated with early-persistent
asthma in childhood. Congratulations for your work!
However, the article cannot be published in this form. In the article the methods paragraph appears after the discussion. Please move it before the results.
I recommend that the abstract has the same form as the article: introduction, methods, results, conclusions, marking them in the abstract.
I recommend you to expand the discussions.
In table no. 1, being on 2 pages, there is a gap.
Author Response
|
Reviewer 1 |
|
|
|
In the article the methods paragraph appears after the discussion. Please move it before the results. |
As suggested, we have moved the methods before results and discussion. |
Moved line 242-319 to Page 2-3 line 69-146. |
|
I recommend that the abstract has the same form as the article: introduction, methods, results, conclusions, marking them in the abstract. |
Thank you for the recommendation. We have added the relevant headings in the abstract. |
Page 1. Line 18, 20, 23, 24, 29. |
|
I recommend you to expand the discussions. |
Thank you for the recommendation, we have expanded the discussion on 1) protopathic bias and reverse causation; 2) confounding factor of respiratory infections 3) Effect size. |
See Page 11. Line 291-304, Page 11. Line 308-315. Page 11 280-286. |
|
In table no. 1, being on 2 pages, there is a gap |
We have reduced the gap and made the changes in Table 1. |
Page 5-6. Line 157-158. |
Reviewer 2 Report
Greetings.
1) Please, consider placing "Methods" in front of the "Results" section.
2) Line 313 "...and adjusted for confounders in the analysis..."
Please specify how was this performed.
3) Discussion
There is relatively plenty of papers with the positive association of early-life ATB and Asthma. In this study, the novelty could be seen in long-term follow up until 6-15 years of age. However, there has been studies suggesting the asociation is not a causal one. Authors have to carefully adress those arguments before any conclusions could be made:
Murk W. Prenatal or early-life exposure to antibiotics and risk of childhood asthma: a systematic review. Pediatrics 2011. "...protopathic bias seem to be possible confounder..."
and
Örtqvist AK. Antibiotics in fetal and early life and subsequent childhood asthma: nationwide population based study with siblin analysis. BMJ 2014:
"in siblings the OR for ATB used in respiratory infections were 2,3, whereas in urinary infections and skin infections treated with antibiotics, there was no difference"
4) Conclusion
In the lines 62-64 the authors state the study investigates 1) asthma trajectories and 2) association ATB-AB. But in the "Conclusion" section, there is just the latter.
Author Response
|
Reviewer 2 |
|
|
|
Please, consider placing "Methods" in front of the "Results" section. |
Thank you for the recommendation. We have moved the methods before results and discussion. |
Moved line 242-319 to Page 2-3 line 69-146. |
|
Line 313 "...and adjusted for confounders in the analysis..."
Please specify how was this performed. |
We have added the following details. “We selected always-low risk asthma group as the reference group and adjusted for confounders in a multinomial regression model”. |
Page 3. Line 140-142 |
|
There is relatively plenty of papers with the positive association of early-life ATB and Asthma. In this study, the novelty could be seen in long-term follow up until 6-15 years of age. However, there has been studies suggesting the asociation is not a causal one. Authors have to carefully adress those arguments before any conclusions could be made:
Murk W. Prenatal or early-life exposure to antibiotics and risk of childhood asthma: a systematic review. Pediatrics 2011. "...protopathic bias seem to be possible confounder..."
and
Örtqvist AK. Antibiotics in fetal and early life and subsequent childhood asthma: nationwide population based study with siblin analysis. BMJ 2014: "in siblings the OR for ATB used in respiratory infections were 2,3, whereas in urinary infections and skin infections treated with antibiotics, there was no difference" |
Thank you for your comments and we agree that there are studies on antibiotics exposure which may not suggest a causal association. We now have added reference to the papers by Murk W et al. and Örtqvist AK et al. in our discussion section and carefully addressed the differences between these prior papers and our own, and the limitations of this work. “Studies have suggested that the association between early-life antibiotic use and childhood asthma could be due to protopathic bias and reverse causation [40]. Protopathic bias occurs when the early signs of disease (which is yet to be diagnosed) is treated with a medication [42]. This might be a potential issue in our study, particularly for the early onset persistent asthma group, as we do not have data available to exclude this possibility. Örtqvist et al [8] have concluded in their study that association between childhood antibiotics and asthma could be impacted by protopathic bias (a form of reverse causation), as the odds for asthma was higher for children treated with antibiotics indicated for respiratory illnesses (which may have been early onset asthma) than urinary or skin conditions. Unlike our study, in their study, early wheezing was not excluded, which could allow protopathic bias to remain. In our study, asthma was assessed from 6-15 years of age, which satisfies the 2-year “wash-out” period criteria between antibiotic exposure and asthma diagnosis. Moreover, respiratory illness in Örtqvist study was assumed by antibiotics categories which may not reflect child conditions correctly. “Murk et al [40] showed that the odd ration drop from 1.38 to 1.16 for the child antibiotics exposure and asthma but remained significant after adjusted respiratory infections, which indicated that there might be potential confounding factors like respiratory infections. However, we have adjusted for potential infections which indicate antibiotic use (hospitalization for fever/viral infection at age 0-1; ear infection or other infectious condition (other than diarrhea/colitis, ear infection and other illnesses), suggesting likelihood bias by infections were reduced. With increasingly unnecessary antibiotics use in children, since majority of respiratory infections for young children are viral that requires no antibiotics use, which could contribute not only to asthma but the global crisis of antibiotics resistance[42, 43].” |
Page 11. Line 309-315; Page 11-12, Line 391-304 |
|
In the lines 62-64 the authors state the study investigates 1) asthma trajectories and 2) association ATB-AB. But in the "Conclusion" section, there is just the latter. |
Thank you, we have added conclusive finding regarding asthma trajectories. “Within LSAC B cohort, we have identified four childhood asthma trajectories including always-low risk, early-resolving, early-persistent, and late-onset asthma.” |
Page 12. Line 350-355 |
Reviewer 3 Report
This is obviously a very important topic, and the authors have access to a very interesting data-set and have provided interesting analyses although there is room for several improvements.
The causal link between antibiotics use and “asthma” is difficult to prove but totally reasonable and possible but not yet certain. The introduction however should be much more clearly worded: what is meant by a “association” (l41), “link” (l46) etc.. The language “Ecological and secular trend evidence” as being an indication for causality should be accompanied by its possible limitations.
What was the testing strategy? Were the questions predefined or was there a search until significant result was found? Has there been an attempt to adjust for multiple testing? This would greatly enhance the strength of the results. In addition, the demographic data of the healthcare card users should also be presented separately.
The results of the analysis with or without healthcare card could be presented side by side. And analysis of all the children without parents with a healthcare card should also be performed.
The wording in the discussion is quite speculative and as in the introduction quite one-sided. (Line203)
There should be a discussion about the effect size of the RR in the statistically significant examples.
The conclusion should be worded more precisely
Minor points
The abstract contains a pointer about antibiotics use against viral infections, but this is not discussed in the main text.
The introduction starts of with Australian dollars without first introducing the fact that the study is about Australia data.
Some of the data in the table has been rounded in an inconsistent fashion
e.g; Table S1 no antibiotics
Author Response
|
Reviewer 3 |
|
|
|
The introduction however should be much more clearly worded: what is meant by a “association” (l41), “link” (l46) etc.. |
Thanks, we have replaced “association (Line 41)” with “This is of concern because antibiotic utilization has been demonstrated to contribute to antibiotic resistance, while also being associated with altered immune maturation, neurodevelopmental disorders, atopic diseases and metabolic disorders in childhood [3].” Link (line 46) “Several cohort studies have now shown that antibiotic exposure under age 1 year is as-sociated with increased risk of childhood asthma up to age 4 years [4-7], especially broad-spectrum antibiotic classes.” |
Page 1. Line 41-43; Page 2. Line 51-52. |
|
The language “Ecological and secular trend evidence” as being an indication for causality should be accompanied by its possible limitations. |
Thanks, we have revised to “Ecological and secular trend evidence indicate that the rate of increase in rates of both antibiotic use and asthma prevalence have risen in parallel, although children were only followed until age 5 years and potential unadjusted confounders [4, 5].” |
Page 2. Line 48-50. |
|
What was the testing strategy? Were the questions predefined or was there a search until significant result was found? Has there been an attempt to adjust for multiple testing? This would greatly enhance the strength of the results. |
Research questions and objectives were predefined at the start of this secondary data analysis. We chose cut off age 6 to exclude transient viral wheezing, which is common in young children. Several subgroup analyses were conducted to test the robustness of the findings. Corrections for multiple comparisons may not be needed as we only compare the early-persistent asthma, late-onset asthma, and early-resolving asthma groups with the always-low risk in one association analysis. |
Page 2 66-68; Page 4. 146-150. |
|
In addition, the demographic data of the healthcare card users should also be presented separately. |
Thanks, we have now added the demographic data for children whose parents have health care cards in Table 1. |
Page 5-6. Line 157-158 (Table 1) |
|
The results of the analysis with or without healthcare card could be presented side by side. And analysis of all the children without parents with a healthcare card should also be performed. |
Thank you, we have made changes in figure 3 and we have added children whose parents without healthcare card. |
Page 9. Line 304-212. Figure 3. |
|
The wording in the discussion is quite speculative and as in the introduction quite one-sided. (Line203) |
Thank you. We have reworded in introduction and discussion, for discussion we have added a paragraph on protopathic bias (reverse causation). For the line 203 we now have added “further studies are required to confirm the proposed mechanism”. |
Introduction: Page 2, 48-56. Discussion: Page 11 280-286. Page 11. Line 289-290. |
|
There should be a discussion about the effect size of the RR in the statistically significant examples. |
Thank you and we have added in the discussion “Our finding of childhood antibiotic exposure associated with a 2-fold increase in risk of early-persistent asthma is consistent with a potentially important effect for children health. Our finding is comparable to the finding of another study, that showed a 1.90-fold increase in odds of current asthma at age 12 years following antibiotic consumption within the first week of life [39]. Furthermore, children exposed to antibiotics are associated with persistent asthma at age 15 -years, which is also likely to persist further into adulthood [18].” |
Page 11. Line 280-285. |
|
The conclusion should be worded more precisely |
Thanks. We have revised as below “Within LSAC B cohort, we have identified four childhood asthma trajectories including always-low risk, early-resolving, early-persistent, and late-onset asthma. Antibiotic exposure between 0-24 months and using more than 3 courses were both associated with increased risk for childhood early-persistent asthma. Our findings emphasize the importance of antibiotic stewardship and greater caution relating to widespread and potentially indiscriminate use of antibiotics in early-life” |
Page 12. Line 350-355 |
|
The abstract contains a pointer about antibiotics use against viral infections, but this is not discussed in the main text. |
Thanks for pointing this out. We have now added “The majority of respiratory infections experienced by young children are viral in nature, and do not require antibiotics treatment. Unnecessary treatment of viral infections with antibiotics contributes to the global crisis of antibiotics resistance [42, 43].” in the discussion. |
Page 12. Line 315-318. |
|
The introduction starts of with Australian dollars without first introducing the fact that the study is about Australia data. |
Thanks and we have now added context for Australia. “…in Australia” |
Page 1. Line 36 |
|
Some of the data in the table has been rounded in an inconsistent fashion
e.g; Table S1 no antibiotics |
Thank you. We have fixed rounding error. |
Supplementary materials (Table S1) |
Reviewer 4 Report
The authors applied group-based trajectory modeling to explore asthma risk factors and factors that affect the asthma phenotype in children. Among the factors explored were the age of antibiotic exposure, the number of antibiotic courses received, sex, antibiotic class and the respective interactions. Confounders were also took into consideration and the analysis was adjusted for factors such as maternal smoking, maternal alcohol, Socio-Economic Indexes for Areas (SEIFA), low birthweight, delivery mode, preterm birth etc. The analysis showed that early-life antibiotic exposure is associated with increased risk of early-persistent childhood asthma, while second-generation cephalosporins seem to increase the risk the most.
Overall the article is very well written, well organized and the analysis well performed. The results are also very interesting for pediatric clinical practice. I have no further comments.
Author Response
|
Reviewer 4 |
|
|
|
Overall the article is very well written, well organized and the analysis well performed. The results are also very interesting for paediatric clinical practice. I have no further comments. |
Thanks very much, no changes. |
NA |
Round 2
Reviewer 3 Report
The changes in the manuscript are adequate, i have no further comments.